# Quinolone Resistance in *Gallibacterium anatis* Determined by Mutations in Quinolone Resistance-Determining Region

**DOI:** 10.3390/antibiotics12050903

**Published:** 2023-05-13

**Authors:** Kasper Rømer Villumsen, Toloe Allahghadry, Magdalena Karwańska, Joachim Frey, Anders Miki Bojesen

**Affiliations:** 1Department of Veterinary and Animal Sciences, University of Copenhagen, 1870 Copenhagen, Denmark; toloe@sund.ku.dk; 2Department of Epizootiology with Exotic Animal and Bird Clinic, Wroclaw University of Environmental and Life Sciences, 50-366 Wrocław, Poland; magdalena.karwanska@upwr.edu.pl; 3Vetsuisse Faculty, University of Bern, 3012 Bern, Switzerland; joachim.frey@unibe.ch

**Keywords:** antimicrobial resistance, fluoroquinolone, *Gallibacterium anatis*, poultry, quinolone, quinolone-resistance-determining region

## Abstract

Control of the important pathogen, *Gallibacterium anatis*, which causes salpingitis and peritonitis in poultry, relies on treatment using antimicrobial compounds. Among these, quinolones and fluoroquinolones have been used extensively, leading to a rise in the prevalence of resistant strains. The molecular mechanisms leading to quinolone resistance, however, have not previously been described for *G. anatis*, which is the aim of this study. The present study combines phenotypic antimicrobial resistance data with genomic sequence data from a collection of *G. anatis* strains isolated from avian hosts between 1979 and 2020. Minimum inhibitory concentrations were determined for nalidixic acid, as well as for enrofloxacin for each included strain. In silico analyses included genome-wide queries for genes known to convey resistance towards quinolones, identification of variable positions in the primary structure of quinolone protein targets and structural prediction models. No resistance genes known to confer resistance to quinolones were identified. Yet, a total of nine positions in the quinolone target protein subunits (GyrA, GyrB, ParC and ParE) displayed substantial variation and were further analyzed. By combining variation patterns with observed resistance patterns, positions 83 and 87 in GyrA, as well as position 88 in ParC, appeared to be linked to increased resistance towards both quinolones included. As no notable differences in tertiary structure were observed between subunits of resistant and sensitive strains, the mechanism behind the observed resistance is likely due to subtle shifts in amino acid side chain properties.

## 1. Introduction

*Gallibacterium anatis* (*G. anatis*) is a member of the *Pasteurellaceae* family and a common inhabitant of the upper respiratory tract and the lower reproductive tract of various bird species, including chickens, turkeys, ducks and geese [1,2,3]. *G. anatis* is a potential pathogen particularly in egg-laying chickens where lesions such as salpingitis and peritonitis are the most frequently reported [4,5]. The main intervention against *G. anatis*-induced disease is antimicrobial treatment; yet several studies have demonstrated a remarkably high prevalence of multidrug-resistant strains [6,7,8,9,10]. The antimicrobial resistances comprise several antimicrobial classes, with tetracycline and sulfonamide-trimethoprim being the most frequent [6,9]. In general, the frequency of resistance determinants in *G. anatis* appears to be higher than in the related organisms *Pasteurella multocida* and *Mannheimia haemolytica* [11]. In particular, resistance against quinolones and fluoroquinolones seems to be relatively common. Quinolones are broad-spectrum antimicrobials, and, in particular, fluoroquinolones such as enrofloxacin (ENR) have become very popular since their introduction for veterinary use in the late 1980s. Unfortunately, widespread use has sparked widespread selection of resistant strains. From a collection of 46 *G. anatis* isolates originated from Mexico and Denmark, the fractions of nalidixic acid (NAL) and ciprofloxacin resistant isolates were 65% and 54%, respectively [6]. More recently, an investigation of a collection of 21 Iranian *G. anatis* isolates revealed 90.5% of the strains resistant [10].

To our knowledge, no previous reports have been published on the molecular mechanisms promoting quinolone resistance in *G. anatis*. Quinolones are broad-spectrum antimicrobial agents inhibiting bacterial DNA replication through interaction with the DNA gyrase and/or topoisomerase IV enzymes. Resistance typically arises from mutations in the quinolone resistance-determining regions (QRDRs) but may also depend on active efflux or protection of the enzymes by Qnr proteins [12]. The aim of the current investigation was to characterize the molecular background for variable susceptibility to quinolones in a collection of 76 well-characterized *G. anatis* strains from 5 different countries isolated over a 41-year period.

## 2. Results

### 2.1. Quinolone Resistance—In Vitro

Minimum inhibitory concentrations (MIC) towards NAL, as well as ENR for each isolate, are shown in Appendix A, and their distribution and development over time is summarized in Figure 1.

Keeping in mind the limited number of available binning intervals and the resulting compression of the MIC value data stemming from the lower-than/greater-than bins, bimodal distributions were observed for MIC values toward both NAL and ENR from their respective density plots (Figure 1A,C). A unimodality test ended with the rejection of the null-hypothesis that the frequency distributions were unimodal (*p* < 2.2 × 10^−16^). Assuming bimodal distributions, modes at 5.6 and 64 µg/mL were observed for NAL, and modes at 0.14 and 2 µg/mL were observed for ENR, suggesting distinct populations of sensitive and resistant isolates. To the best of our knowledge, no clinical breakpoints for nalidixic acid exist for *G. anatis*. For ENR, however, the Clinical and Laboratory Standards Institute (CLSI) VET06 supplement state MIC ≤0.25 µg/mL as sensitive, 0.5–1 µg/mL as intermediate and ≥2 µg/mL as resistant [13]. These breakpoints are in line with the observed bimodal distribution for ENR in the present study.

When plotting the observed MIC value of each strain relative to the year of isolation, there was an increased observed tolerance to both NAL and ENR over time, suggesting a development of resistance (Figure 1B,D).

### 2.2. Quinolone Resistance—In Silico

In silico determination of antimicrobial resistance (AMR) profile was performed using the ResFinder database. Specifically for quinolones, the ResFinder query covered *qnr* genes and efflux pumps (*qep* and *oqxAB* family genes), as well as cyclic adenosine monophosphate receptor proteins (crp family genes) and aminoglycoside acetyltransferases (*aac(6′)* family genes) capable of impeding the effect of quinolone treatment. The searches identified AMR genes, predominantly relating to tetracycline and beta lactam categories; however, across all 76 included isolates, no AMR genes related to the included quinolones (“ciprofloxacin”, “fluoroquinolone”, “nalidixic acid”, “unknown quinolone”) were observed.

### 2.3. Genetic Determinants

To investigate potential effects of point mutations on subunits of the quinolone targets DNA gyrase (*gyrA* and *gyrB*), as well as topoisomerase IV (*parC* and *parE*) genes, translated amino acid sequences of each gene were extracted and sequences from each isolate were aligned. During alignment of ParC amino acid sequences, an 11-amino acid N-terminal tail was observed for 21 of the 76 sequences. As the positions mentioned in the following text refer to consensus sequence positions, note that these positions will be shifted for the remaining 55 sequences. Appendix A includes the isolate-specific amino acid identified for each position that displayed ≤70% (or >30% variation) for each of the four genes. This cutoff threshold was chosen to limit the number of potential variants to those displaying substantial variation in order to relate the desired extent of observed variation to the observed MIC value frequency distributions. A total of nine positions across the four subunits were found to satisfy the threshold for variation. Figure 2 summarizes the results as the development, or mutation, in each identified variable position over time.

For GyrA, notable shifts in amino acid prevalence were observed. For position 83 a shift away from serine (S) was observed over time, while alternatives to aspartate (D) were emerging for position 87. Concurrent variants were observed for position 426. Position 64 in GyrB showed a somewhat concurrent presence of two variants. In ParC, position 88 was shown to transition from being strictly glutamate (E) in strains isolated prior to 2010, to the occurrence of glycine (G) in particular, and also alanine (A) and lysine (K) alternatives in strains isolated later than 2010. For ParC position 205, an early valine (V) variant was observed, while concurrent variants were observed throughout the study period. Position 289 similarly showed concurrent variants, while positions 514 and ParE position 373 seemed to indicate an emergence of alternatives to valine and alanine, respectively.

To consider the observed variation from a resistance perspective, the associations between variation within each identified position and the corresponding MIC values were investigated. When recorded MIC values were plotted according to allele variant for each varying position, seven of the nine variable positions displayed statistically significant variation in one, or both, of the recorded MIC values, as shown in Figure 3, parts 1 and 2.

For NAL, significant differences in MIC values were observed for the following gene products and positions: GyrA pos. 83, 87 and 426; ParC pos. 205 and 289; and ParE pos. 373.

For ENR, significantly different MIC values were observed for: GyrA pos. 83 and 87; and for ParC pos. 205, 289 and 514.

To gain further insight into the context, or relative importance, of the individual variable sites, a concatenated string of amino acids was constructed for each isolate, combining the one-letter amino acid abbreviations for the nine variable sites. A total of 40 different concatenated strings were formed from the 73 isolates with both sequence and MIC value information, as shown in Figure 4.

From Figure 4, some variation was observed in terms of NAL MIC values; however, together with the ENR MIC values, a general pattern of low MIC values is found for concatenated profiles of the general structure SDXXEXXXX (serine for GyrA position 83, aspartic acid for GyrA position 87, variables for GyrA position 426 and GyrB 64, glutamate in ParC position 88 and variable amino acids for the remaining ParC and ParE positions). In addition to this, other notable patterns were observed. In the present dataset, a serine in position 83 of GyrA only occurred together with an aspartic acid in position 87. If position 83 was shifted to one of the variants, however, higher MIC values were observed, despite an aspartic acid in position 87, further underlining the importance of GyrA position 83. Furthermore, for ParC position 514, valine variants were associated with low as well as high MIC values for both NAL and ENR, when seen in isolation. The concatenated strings, however, show that a valine is present in all but one of the SDXXEXXXX-type strings associated with low MIC-values, as well as several strings associated with high MIC values, decreasing the relative impact of this position on resistance phenotype, despite the significant difference in MIC values observed in isolation.

When combining the statistically significant differences in MIC values for isolated positions, the shifting prevalence of amino acids in the various positions, as well in the concatenated strings, variation in GyrA 83 + 87 and ParC 88 indicate a process in which single amino acid variants, prior to the year 2000, transition to multiple alternative variants post-2000, all marking a general shift towards the higher MIC levels.

### 2.4. Implications of Mutations on Protein Structure and Resistance

To evaluate the data describing amino acid prevalence over time, and the MIC value as a function of variation in the identified key positions in a structural context, structural prediction models were created for each protein subunit for a low-MIC representative (10672-9), as well as a high-MIC representative (80-02) strain. Revolving models of GyrA, ParC and ParE are included as Appendix A for further exploration.

#### 2.4.1. GyrA

Three key positions were observed in GyrA. To further investigate the effects of changes in primary structure, their potential effects on tertiary structure and function in terms of susceptibility to quinolones, a further examination of the structural changes was performed.

For position 83 in the primary structure, a shift over time from a serine (S) residue with a polar, yet neutral, side chain, to tyrosine, matching the polar, neutral side chain characteristic of serine, or non-polar and neutral phenylalanine (F), valine (V) or leucine (L) residues was associated with increased tolerance towards both NAL and ENR. These three variants are all hydrophobic relative to serine. For position 87, aspartic acid (D, polar, negative, hydrophilic) was observed throughout the timeframe of this study, with alanine (A, non-polar, neutral, hydrophobic), asparagine (N, polar, neutral, hydrophilic), tyrosine (Y, polar, neutral, hydrophobic) and glutamic acid (E, polar, negative, hydrophilic) appearing as alternative alleles. While the shift from negative to neutral side chain was associated with increased MIC values, aspartic acid and glutamic acid, both with polar, negative, hydrophilic side chains, were associated with mixed or lower NAL MIC values. Finally, position 426 displaying concurrent alanine (A, non-polar, neutral, hydrophobic) and threonine (T, polar, neutral, neutral hydrophobicity) variants showed significantly lower NAL MIC values for the threonine variant.

Structural prediction of GyrA from isolates 10672-9 and 80-02, chosen as representatives of low- and high-MIC isolates, respectively, suggests that positions 83 and 87 sit one turn apart within an alpha-helix, in the same orientation, consistent with the distance between them in the primary structure, as shown in Figure 5. As for positions 83 and 87, position 426 was also shown to be part of an alpha-helical structure. When superimposed, no apparent difference in local secondary or tertiary structure between GyrA from 10672-9 and 80-02 were observed, suggesting physiochemical changes in side chain structure as potential mechanistic explanations for altered quinolone sensitivity.

#### 2.4.2. GyrB

No significant changes in MIC value were observed for variants in GyrB pos. 64.

#### 2.4.3. ParC

Within ParC, variations in positions 205, 289 and 514 were associated with significantly different MIC phenotypes. For position 205, threonine (T, polar, neutral, neutral hydrophobicity) and alanine (A, non-polar, neutral, hydrophobic) variants were observed throughout the study timeframe, while a valine (V, non-polar, neutral, very hydrophobic) variant was observed in some of the earliest samples in the present study. These early valine variants are associated with MIC values for both NAL and ENR in the lowest bin, as shown in Figure 3, part 2.

Position 289 displayed concurrent valine (V, non-polar, neutral, very hydrophobic) and isoleucine (I, non-polar, neutral, very hydrophobic) variants. Very similar in structure, the difference in terms of association to MIC values lay in a minor, yet significantly larger fraction of low ENR MIC values for the isoleucine variant.

For position 514, valine (V, non-polar, neutral, very hydrophobic) was observed throughout the study timeframe, while methionine (M, non-polar, neutral, very hydrophobic) appears as a conservative substitution variant in very early isolates, and then again in the later stages of the study period. Valine was associated with the highest fraction of low MIC values for both NAL and ENR.

When observing the predicted models of ParC from 10672-9 and 80-02, the 11-amino acid N-terminal tail is visible for the 10672-9 ParC structure (Figure 6). Consensus position 205 was in a linker region between two alpha-helices, position was 289 within a beta-sheet and position 514 sat as the first amino acid of a downstream beta-sheet.

#### 2.4.4. ParE

In position 373 of the primary structure, an alanine (A, non-polar, neutral, hydrophobic) variant was observed consistently throughout the study period, while a valine (V, non-polar, neutral, very hydrophobic) was observed early and then again late, and a threonine (T, polar, neutral, neutral hydrophobicity) variant occurred only in later isolates. Significant differences in MIC values were only observed for NAL, where both the threonine variant and the valine variant appear to mark a shift towards higher MIC values. Upon superimposition, a slight offset between the 10672-9 and 80-02 predictions were observed, however, with position 373 placed identically within an alpha-helical secondary structure (Figure 7).

## 3. Discussion

Our aim was to identify the molecular basis for quinolone resistance in *G. anatis* by combining phenotypic data, sequence data and structural prediction. MIC values for both NAL and ENR were generated, revealing distinct, bimodal frequency distributions of both sets of MIC values. While no official clinical breakpoints for NAL exist for *G. anatis*, the recorded MIC values generally separate the isolates in the present study into low- and high-MIC categories, with a small intermediate category for both NAL and ENR, supported for ENR by CLSI breakpoints. By far, the majority of isolates fell into high-MIC categories, demonstrating a notable, general decreased sensitivity, or improved resistance, to NAL and ENR in these isolates.

To connect the observed resistant phenotypes to molecular traits, sequence data for each isolate were investigated to identify potential mechanisms or traits within the genomes. Quinolone resistance mechanisms have previously been divided into separate classes: chromosome-, plasmid-, or target-mediated quinolone resistance, as reviewed by Aldred, et al. [14]. While chromosome-mediated resistance mechanisms relate to modulation of the net-intake of quinolone by altering in- and efflux through porins or pumps, plasmid-mediated resistance mechanisms relate to plasmid-borne genes encoding quinolone antagonists (*qnr*-genes), efflux pumps or enzymes that can alter specific quinolone types and thus decrease their efficacy [14,15]. For *E. coli* strains isolated from animals, including duck and goose, *qnr*-genes, as well as *aac*(6′)-genes, have previously been observed along with mutations in the QRDR [16]. Based on the available sequence data for the present study, however, no quinolone-related resistance genes were identified in any of the 76 *G. anatis* isolates, whether plasmid-borne or chromosomal. Whether this might potentially be due to loss of plasmid DNA during propagation of the strains or during DNA isolation is difficult to assess due to the archival nature of many of the strains included in this project. Of the most recently sequenced strains in this project, in silico searches for mobile genetic elements identified a plasmid in just one of the forty isolates from Poland; however, whether this limited presence of plasmids is instructive in terms of the remaining isolates is purely speculative [17]. It is worth including, however, that to the best of our knowledge, *qnr*, *qep* and *oqxAG* genes have not been observed for *Pasteurellaceae*, which would be in line with our current findings.

Target-mediated resistance mechanisms refer to specific mutations in DNA gyrase and topoisomerase IV genes that inhibit quinolone binding to their protein subunits targets. In the present project, we identified nine positions within DNA gyrase and topoisomerase IV subunit primary structures that exhibited substantial (>30%) variation. This variation in local protein structure was then connected to statistically significant variations in phenotypic MIC values for NAL, ENR or both for seven of those positions. The overall, gradual increase in MIC values over time observed in Figure 1 was supported by an increase in variant diversity over time for GyrA positions 83 and 87, as well as ParC position 88. Variations in these three positions were associated with changing MIC values, both seen in isolation for the two GyrA positions, but in particular when considered in the context of all identified, variable sites.

When observing positional variation in a structural context, no apparent tertiary structure changes in any of the modelled subunits were observed between the structural model predictions made for the low- and high-MIC value representatives 10672-9 and 80-02, respectively. This emphasizes the importance of the amino acids themselves, including their respective side chains, rather than structural changes in secondary or tertiary structure. Considering the impact of GyrA positions 83 and 87 observed from both the isolated positional MIC variations, as well as their relative importance seen from the concatenated string variants, it is notable that they were predicted to sit one turn apart, facing the same direction in an alpha helical structure, consistent with predicted sites of interaction with quinolones, as previously reviewed [14,15]. A similar pattern has been described for multiple other bacterial pathogens, including *E. coli* and *Klebsiella pneumoniae* [18,19]. In a recent report, mutation from a serine in GyrA position 81 was demonstrated in quinolone-resistant *Streptococcus agalactiae*, likely a similar function to that of position 83 in the present study, although shifted in the primary structure [20]. For studies such as this, a larger dataset would always be preferable. Perhaps future studies will benefit from an increased availability of sequence data with appropriate metadata on phenotypic traits, such as antimicrobial resistance patterns. Furthermore, targeted mutagenesis, while beyond the scope of the present study, could provide valuable mechanistic validation of the present findings. However, given the drug mechanism of action, the timeline and the structural data from this study, as well as previous studies, and the described changes in side chain physiochemical properties, local changes in the primary structure for GyrA and ParC could provide a plausible, mechanistic explanation for increased quinolone resistance in *G. anatis*, likely accelerated through short-term evolutionary pressures of past and current antibiotic treatment plans [17].

In conclusion, a combined analysis of phenotypic resistance and genomic sequence data for 76 isolates of *G. anatis*, spanning four decades, suggest a general acquisition of resistance towards NAL and ENR over time through a target-mediated resistance mechanism. As no acquired resistance genes were observed, mutations of quinolone targets within the QRDR, and particularly in GyrA and ParC, provide a plausible mechanism of resistance, similar to previously published mechanisms for other bacterial pathogens, a development likely enforced by evolutionary pressure from antimicrobial treatment regimens.

## 4. Materials and Methods

### 4.1. Bacterial Strains

A total of 76 previously published strains were included in the present study. All 76 strains had been sequenced previously (see Appendix A for accession information). Strains were selected to represent a maximized time interval, broad geographic distribution and varying levels of susceptibility to nalidixic acid and enrofloxacin [6]. All strains were isolated from chickens, geese and, in one case, from a duck, during the period 1979 to 2020 [1,4,6,17,21] (Appendix A). Forty isolates originated from Poland (AGRO-VET Veterinary Laboratory in Wrocław), twenty-one strains originated from Iran, nine from Denmark, three from Germany, two from Mexico and one was from the USA. The strains originated from fundamentally different production systems including battery-cage, free-range and organic systems and cover layers, layer parent stock and broilers.

### 4.2. Minimum Inhibitory Concentration

Of the 76 strains included, MIC against the 1st generation quinolone NAL and the fluoroquinolone ENR were already available for 40 of the strains, based on a microdilution assay [17]. To complete the dataset, corresponding MIC values for the remaining strains were determined using a broth microdilution assay, as described previously [10]. Both novel and previously determined MIC values are given in Appendix A. Finally, to allow for comparison between MIC values determined using different concentration ranges, all MIC values were then integrated into the MIC range sub-interval—or bin—setup used in the present study for further analyses.

### 4.3. Sequence Data

All sequences included were sequenced prior to this study. Origin and accession IDs are given in Appendix A.

### 4.4. Initial QC

All raw sequences available as fastq-files were checked for sequence quality using FastQC (v0.11.9) [22], and the results were summarized using MultiQC (v1.12) [23]. Identified Illumina adapter sequences were removed using AdapterRemoval (v2.3.2) [24] using an Illumina Universal Adapter sequence from the FastQC github repository (https://github.com/golharam/FastQC/blob/master/Configuration/adapter_list.txt (accessed on 9 August 2022)—AGATCGGAAGAG—except for sequences from Polish isolates, where the following adaptor sequences were used: --adaptor1 TCGTCGGCAGCGTCAGATGTGTATAAGAGACAG and –adaptor2 GTCTCGTGGGCTCGGAGATGTGTATAAGAGACAG. We trimmed for quality (sliding window approach, window size = 4, Phred threshold = 20), resulting in trimmed sequences.

### 4.5. Bioinformatic Analyses

Trimmed sequences were submitted to the online version of ResFinder 4.1 (https://cge.food.dtu.dk/services/ResFinder/, accessed on 14 February 2023), selecting “Other” for the species selection, enabling the search for acquired antimicrobial genes [25,26,27]. For the type strain NCTC11413 and strain UMN179, only assembled contigs were available and these were submitted as contigs. A collected table of ResFinder output has been included as Appendix A. Draft genome assemblies were made using the trimmed sequences using MEGAHIT (v1.2.9) with default settings [28]. Subsequently assemblies were assessed and summarized using QUAST (see Appendix A) [29]. Each assembly was then annotated using Prokka (v1.14.6) using default settings, but with genetic code set to 11 [30].

Translated amino acid sequences of the *gyrA*, *gyrB*, *parC* and *parE* genes were retrieved from each annotated genome using Artemis (v18.1.0), and sequences for each gene were aligned using Clustal Omega (v1.2.4) [31,32]. Alignments were then visualized and compared using Jalview (v2.11.2.5) [33]. An arbitrary threshold of ≤70% consensus for each position was set to identify variable positions with which to proceed. Positions refer to the consensus position within the multiple sequence alignment for each of the four subunits. Amino acids for each of the identified variable positions were recorded along with the existing metadata for each strain (see Appendix A), and the resulting dataset was then analyzed using R (v4.2.1, “Funny-Looking Kid”) via RStudio (v. 2022.12.0+353) [34]. The following packages were used during the analyses: readxl [35], ggplot2 [36], dplyr [37], forcats [38], multimode [39] and ggpubr [40]. Structural predictions of protein subunits were made using AlphaFold (v2.1.0) via the Google Colab notebook using the monomer mode and the run_relax flag option and subsequently visualized and annotated using ChimeraX (v1.5rc202210282349) [41,42,43].

### 4.6. Statistical Analyses

All statistical analyses address the null hypothesis that the groups compared belong to the same underlying data distribution, while the alternative hypothesis is that data for each group do not belong to the same underlying data distribution and that the data are significantly different. All statistical analyses are performed using R through RStudio (see previous section). The test for unimodality was conducted using the multimode r package. As MIC values relative to specific mutations were not expected to be normally distributed, and as the binning of MIC values results in discrete data groups, non-parametric tests were used. All other tests are identified within the relevant figures.

## Figures and Tables

**Figure 1 antibiotics-12-00903-f001:**
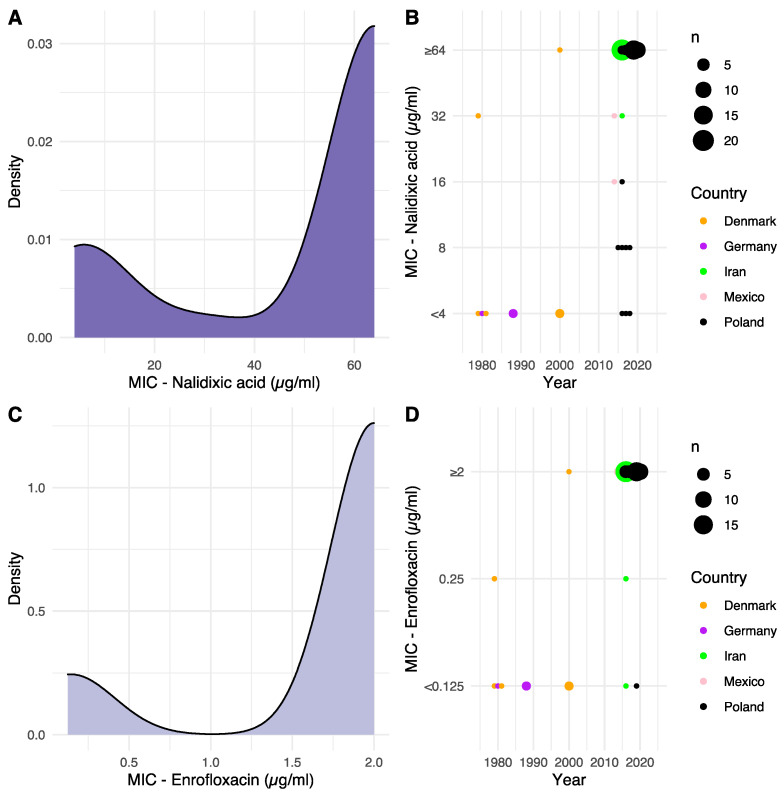
Distribution of isolate MIC and development over time. (**A**,**C**) Density plot distribution of MIC values against NAL and ENR, respectively. (**B**,**D**) Determined MIC values, according to year of isolation, against NAL and ENR, respectively. Dot size is relative to the number of isolates, and dot color indicates the geographical origin of the isolates. For the two Mexican strains, the date of submission to NCBI was used, as no date of isolation was available.

**Figure 2 antibiotics-12-00903-f002:**
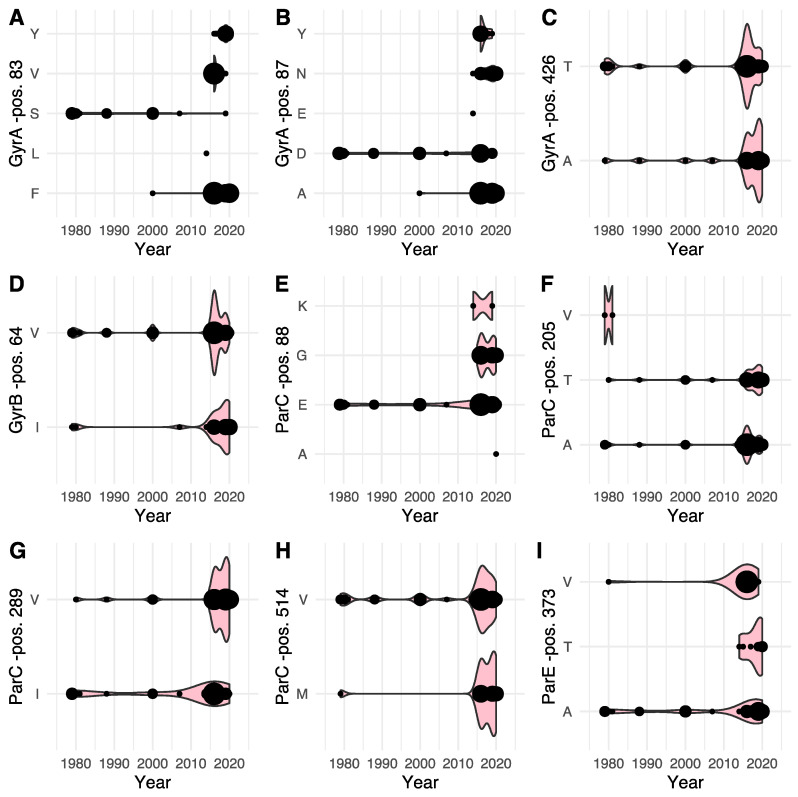
Combined dot and violin plot (pink) of the prevalence of specific amino acids at each identified variable position within the GyrA (**A**–**C**), GyrB (**D**), ParC (**E**–**H**) and ParE gene (**I**) according to year of isolation. The size of each dot, as well as the width of the violin plot is relative, increasing with the number of observations at a given time point.

**Figure 3 antibiotics-12-00903-f003:**
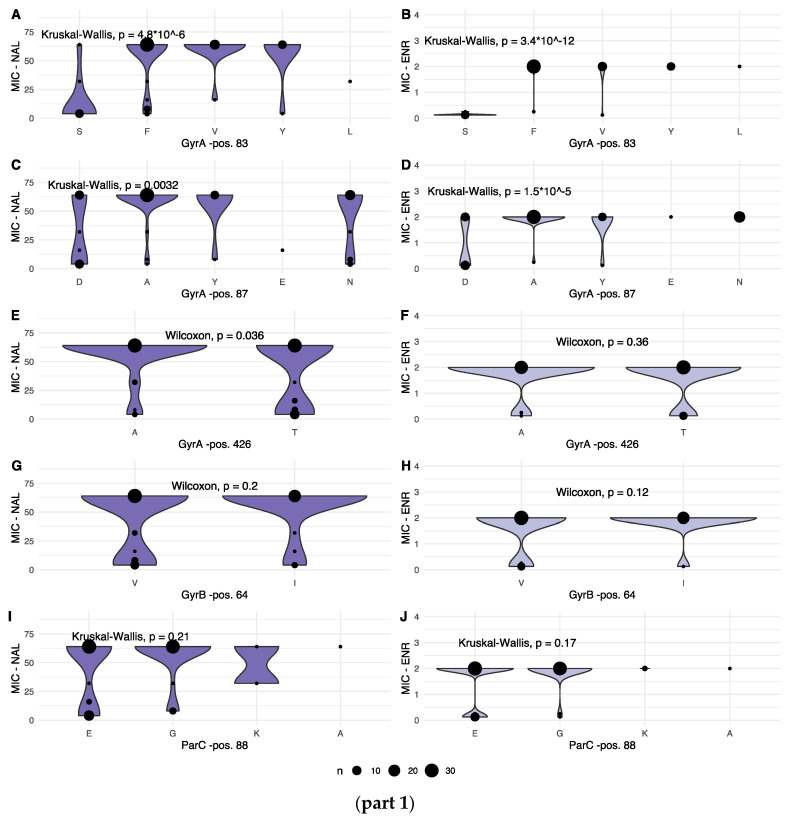
(**part 1**): Combined dot and violin plot showing the distribution of MIC values for NAL (left column) and ENR (right column) for each amino acid in variable positions in GyrA (**A**–**F**), GyrB (**G**,**H**) and ParC genes (**I**,**J**). Details of each statistical test are given for each panel. The size of each dot, as well as the width of the violin plot, is relative, increasing with the number of observations of a given MIC. (**part 2**): Combined dot and violin plot showing the distribution of MIC values for NAL (left column) and ENR (right column) for each amino acid in variable positions in the ParC (**K**–**P**) and ParE gene products (**Q**,**R**). Details of each statistical test are given for each panel. The size of each dot, as well as the width of the violin plot is relative, increasing with the number of observations of a given MIC.

**Figure 4 antibiotics-12-00903-f004:**
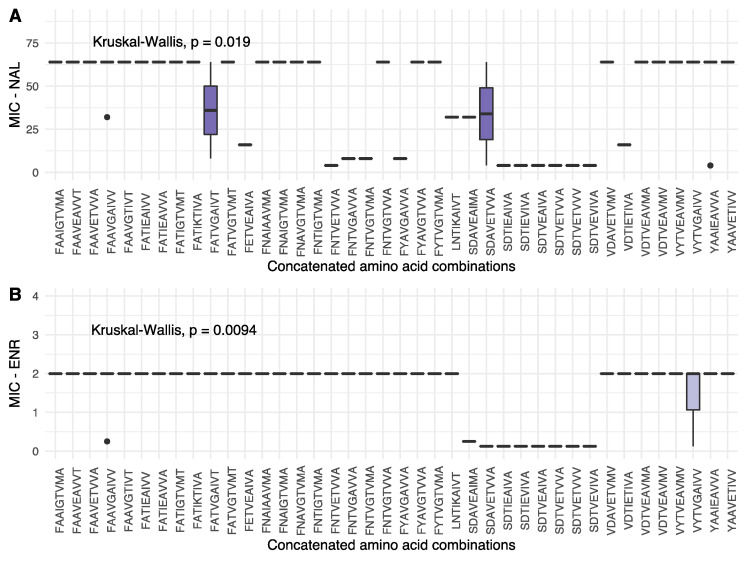
Box plot showing the distribution of MIC values for NAL (**A**) and ENR (**B**) for each concatenated amino acid profile observed in the present study. Each boxplot shows the median as a horizontal bar, and the upper and lower quartiles for each entry. The order of the amino acids in each string follow that of Figure 2: GyrA 83, GyrA 87, GyrA 426, GyrB 64, ParC 88, ParC 205, ParC 289, ParC 514 and ParE 373. Statistical details are given for each figure.

**Figure 5 antibiotics-12-00903-f005:**
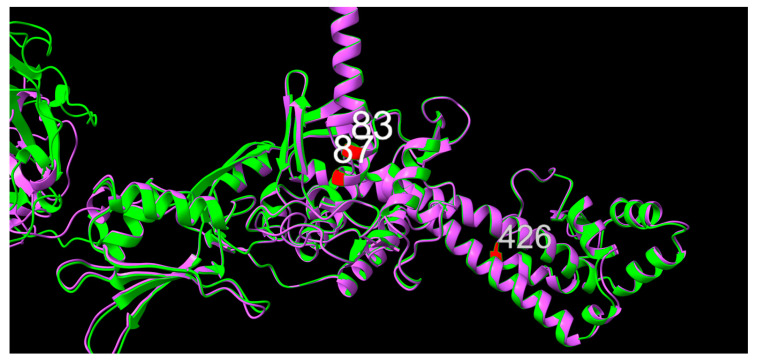
Graphic overlay of structural models of GyrA from (green) 10672-9 and (purple) 80-02. Positions 83, 87 and 426 are shown in red. Full structures are included as Appendix A.

**Figure 6 antibiotics-12-00903-f006:**
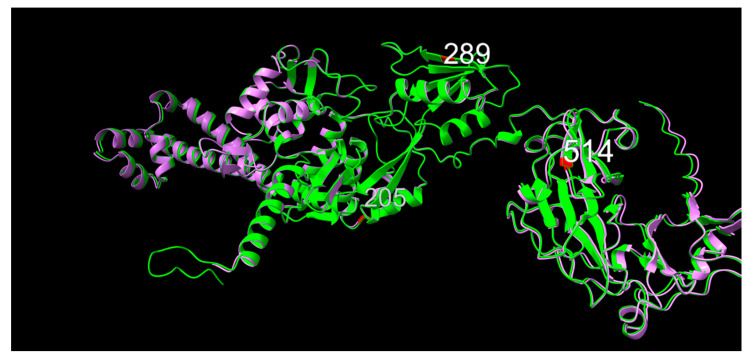
Graphic overlay of structural models of parC from (green) 10672-9 and (purple) 80-02. Positions 205, 289 and 514 are shown in red. Full structures are included as Appendix A.

**Figure 7 antibiotics-12-00903-f007:**
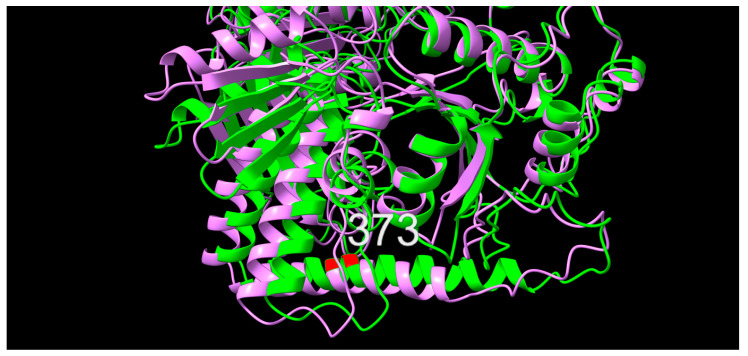
Graphic overlay of structural models of ParE from (green) 10672-9 and (purple) 80-02. Position 373 is shown in red. Full structures are included as Appendix A.

## Data Availability

The datasets supporting the conclusions of this article are provided in the Appendix A.

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
