# Peer review of "Quinolone Resistance in Gallibacterium anatis Determined by Mutations in Quinolone Resistance-Determining Region"

_antibiotics, 2023, doi:10.3390/antibiotics12050903_

Round 1

Reviewer 1 Report

The article on title “Quinolone resistance in Gallibacterium anatis determined by mutations in Quinolone Resistance Determining Region” lays out a very interesting connection between the emergence of resistant bacteria using  phenotypic antimicrobial resistance data with genomic sequence data from a collection of G. anatis strains isolated from poultry between 1979 and 2020. The relevant data show that there is statistically significant when the analysis of  variable positions in the primary structure of quinolone protein targets with antimicrobial resistance are correlationed.  The manuscript is well performed and easy to follow. I think that this provides important results about the knowledge and futures perspectives for microbial detection of G. anatis and evaluation of quinolone resistance genes in poultry. I recommend for acceptance with comments addressed below: 

1.-Abstract is adequately described.

2.- The introduction provide sufficient background and include relevant references.

3.- The methodology is adequately described, however, is confuse the section about the “….addition to these, corresponding MIC values were determined using a broth microdilution assay, as described previously”, it must to be clarify, it was made for all bacterial strain?.

4.- A major discussion is necessary as mutations for quinolone resistance determining region are important for the resistances, prevalence and functions.

Author Response

The article on title “Quinolone resistance in Gallibacterium anatis determined by mutations in Quinolone Resistance Determining Region” lays out a very interesting connection between the emergence of resistant bacteria using  phenotypic antimicrobial resistance data with genomic sequence data from a collection of G. anatis strains isolated from poultry between 1979 and 2020. The relevant data show that there is statistically significant when the analysis of  variable positions in the primary structure of quinolone protein targets with antimicrobial resistance are correlationed.  The manuscript is well performed and easy to follow. I think that this provides important results about the knowledge and futures perspectives for microbial detection of G. anatis and evaluation of quinolone resistance genes in poultry. I recommend for acceptance with comments addressed below:  

1.-Abstract is adequately described.

- We appreciate this observation.

2.- The introduction provide sufficient background and include relevant references. 

­- We are glad to hear this.

3.- The methodology is adequately described, however, is confuse the section about the “….addition to these, corresponding MIC values were determined using a broth microdilution assay, as described previously”, it must to be clarify, it was made for all bacterial strain?. 

- We agree with this comment. The lines in question have now been changed to the following:

“Of the 76 strains included, minimum inhibitory concentrations (MIC) against the 1st generation quinolone nalidixic acid (NAL) and the fluoroquinolone enrofloxacin (ENR) were already available for 40 of the strains, based on a microdilution assay [17]. To complete the dataset, corresponding MIC values for the remaining strains were determined using a broth microdilution assay, as described previously [10].”

4.- A major discussion is necessary as mutations for quinolone resistance determining region are important for the resistances, prevalence and functions.

  • We agree to this point and believe we have balanced the discussion accordingly.

Reviewer 2 Report

1. Brief summary

I value the opportunity to review this interesting manuscript. The aim of the paper entitled “Quinolone resistance in Gallibacterium anatis determined by mutations in Quinolone Resistance Determining Region” is appropriate for consideration of publication in the Antibiotics. The manuscript is moderately structured and written. However, there are major concern about this manuscript.

2. General concept comments

The data reported in the manuscript is scientifically interesting. Unfortunately, only 44.18% (19/43 publications) of recent references published over the previous five years (2018-2023) are cited.Please consider updating the references.

- The unspecific title of this paper is insufficiently reflecting its contents. I am pleased to suggest that the term concerning tested isolates (from which animals, as well as the form study area), might be included.

- Please arrange the keywords alphabetically for a standardized presentation.

- More specifics are required in your section on materials and methods. Please describe the statistics that were used in this study. 

- On results (Quinolone resistance – in vitro) fortested isolates, the MIC distribution and descriptive statistics are important as well. Please consider more details (range, median, and 90th percentile, IQR) to clarify the experimental outcomes.

- Since the study included a variety of poultry husbandries (chickens, geese and duck), different patterns of antibiotic consumption and management strategies were hypothesized. If the results of the experiment are not applied beyond other animal species, more information is required in your discussion.

- Your discussion section needs more detail. Since, the reference 10 (Allahghadry, et al.), published in Veterinary Research in 2021, addresses a content very close to the study currently presented.

- I would advise the author to outline the study's limitation in the discussion section.

- All scientific names should be italicized Gallibacterium anatisG. anatis, Pasteurella multocida”. Please double-check the name and symbol of each gene/protein also.

- The conclusion section of this manuscript is quite vague and lengthy.I recommend that authors answer the objective of the work in the conclusion with accurate information. Some conclusions should state the primary point of the obtained major content and make recommendations for clinical practice. Additionally, this research's conclusion section requires a suggestion for further research.

- Finally, I am grateful for the chance to review this submission. Although the topic is valuable and interesting, further interpretation and motivation is required. Wishing the authors success for their paper publication, I hope the suggestions are beneficial.

Author Response

Reviewer 2

  1. Brief summary

I value the opportunity to review this interesting manuscript. The aim of the paper entitled “Quinolone resistance in Gallibacterium anatis determined by mutations in Quinolone Resistance Determining Region” is appropriate for consideration of publication in the Antibiotics. The manuscript is moderately structured and written. However, there are major concern about this manuscript.

  • We appreciate the enthusiasm.

  1. General concept comments

- The data reported in the manuscript is scientifically interesting. Unfortunately, only 44.18% (19/43 publications) of recent references published over the previous five years (2018-2023) are cited.Please consider updating the references.

- Searching PubMed with the query “gallibacterium anatis” indeed results in 42 hits. However, among these hits, article titles indicate that topics range from vaccine development and the use of essential oils, over antimicrobial resistance and on to post-mortem survival of the pathogen, virulence factors and diagnostics. As this manuscript never intended to provide a full review of this pathogen, we would respectfully prefer to stay focused on resistance towards quinolone and fluoroquinolones.

- The unspecific title of this paper is insufficiently reflecting its contents. I am pleased to suggest that the term concerning tested isolates (from which animals, as well as the form study area), might be included.

- We respectully disagree with this point, as we believe that the goal and content of this manuscript is clear from the current title, without producing an excessively long title.

- Please arrange the keywords alphabetically for a standardized presentation.

- We agree, and this has now been changed in the manuscript (line27)

- More specifics are required in your section on materials and methods. Please describe the statistics that were used in this study.

- In terms of the statistics section, we provide a clear definition of our basis for hypothesis testing, as well as the software used. To improve clarity, we have added a reference to the “bioinformatic analyses” section for information of the exact software packages used. Since the choice of non-parametric analysis depends on the number of data groups compared, we have chosen to include this information directly in each relevant figure, to avoid excessive complexity and potential confusion. These were conscious decisions, and we would prefer to leave them this way, as we believe that we have provided sufficient information to allow statistical interpretation.

- On results (Quinolone resistance – in vitro) fortested isolates, the MIC distribution and descriptive statistics are important as well. Please consider more details (range, median, and 90th percentile, IQR) to clarify the experimental outcomes.

- This is taken as a comment to the presentation of data in figure 1A, specifically. While we agree that descriptive statistics are important in most data presentations, the goal at this point is to illustrate whether we can describe a bimodal distribution if the recorded MIC values. The reason for this being, that it could help us in differentiating between susceptible and resistant strains, with a possible mid-tier of intermediate strains. This would not be possible through range, single median and percentiles. Instead, we have investigated the dual modes observed, and these are reported in the main text. We hope that this reasoning will be accepted.

- Since the study included a variety of poultry husbandries (chickens, geese and duck), different patterns of antibiotic consumption and management strategies were hypothesized. If the results of the experiment are not applied beyond other animal species, more information is required in your discussion.

- We acknowledge the interest for inclusion of additional information regarding use of different antibiotic consumption in the different avian species included. However, comparable data on these aspects are are not available from all countries involved. In Denmark, antimicrobial use is registered at farm and animal level, yet this is not the case in Poland and Iran. Usage patterns are thus of more anecdotal nature and not really regarded suitable for inclusion in the current paper.    

- Your discussion section needs more detail. Since, the reference 10 (Allahghadry, et al.), published in Veterinary Research in 2021, addresses a content very close to the study currently presented.

- We respectfully disagree. The study by Allahghadry et al provides a detailed description of the phylogenetic relations, virulence genes and phenotypic antimicrobial resistance patterns observed in Iranian isolates of G. anatis. In the present study, we work specifically with the molecular mechanisms underlying resistance to quinolones and fluoroquinolones. We are thus, to the best of our knowledge, presenting the first G. anatis-specific insights into the mechanisms behind one of the phenotypic aspects investigated by Allahghadry et al.

- I would advise the author to outline the study's limitation in the discussion section.

- We appreciate this comment, and have modified the final section of the discussion, prior to the conclusions as follows:

“For studies such as this, a larger dataset would always be preferable. Perhaps future studies will benefit from an increased availability of sequence data with appropriate metadata on phenotypic traits, such as antimicrobial resistance patterns. Furthermore, targeted mutagenesis, while beyond the scope of the present study, could provide valuable mechanistic validation of the present findings. However, given the drug mechanism of action, timeline, structural data from this study, as well as previous studies, and the described changes in side chain physiochemical properties, local changes in primary structure for gyrA and parC could account for a plausible, mechanistic explanation of increased quinolone resistance in G. anatis, likely accelerated through short-term evolutionary pressures of past and current antibiotic treatment plans [17].”

- All scientific names should be italicized “Gallibacterium anatis, G. anatis, Pasteurella multocida”. Please double-check the name and symbol of each gene/protein also.

-We obviously agree and have checked and corrected the paper throughout.  

- The conclusion section of this manuscript is quite vague and lengthy.I recommend that authors answer the objective of the work in the conclusion with accurate information. Some conclusions should state the primary point of the obtained major content and make recommendations for clinical practice. Additionally, this research's conclusion section requires a suggestion for further research.

-

 We agree. The final section of the discussion section has been changed to the following:

“In conclusion, a combined analysis of phenotypic resistance and genomic sequence data for 76 isolates of G. anatis spanning four decades, suggest a general acquisition of resistance towards NAL and ENR over time through a target-mediated resistance mechanism. As no acquired resistance genes were observed, mutations of quinolone targets within the QRDR, and particularly in gyrA and parC, provide a plausible mechanism of resistance, similar to previously published mechanisms for other bacterial pathogens, a development likely enforced by evolutionary pressure from antimicrobial treatment regimens.”

Regarding suggestions for future studies, these have now been added in the discussion (lines 342-347).

- Finally, I am grateful for the chance to review this submission. Although the topic is valuable and interesting, further interpretation and motivation is required. Wishing the authors success for their paper publication, I hope the suggestions are beneficial.

- Again, we appreciate the enthusiasm, and are grateful for the constructive feedback provided.

Reviewer 3 Report

Dear authors, the article is interesting because it deals with a topical issue of great relevance to the scientific community. 

I make some general comments and specific suggestions that I hope will improve the manuscript:

Check for typos

All bacterial species must be reported in italic font

Keyword: add Gallibacterium anatis

Line 42 - Enrofloxacin (ENR)

Line 45 - Nalidix (NAL)

Line 60 - Minimum inhibitory concentrations (MIC)

Bacterial strains: some additional information on the selected strains would be appropriate. Are they part of already published research? How were they stored? How was the quality of the isolates ensured?

Which laboratory carried out the pivotal analysis of the research?

Finally, does the study have any limit?

 Regards,
the Reviewer

Author Response

Reviewer 3

Dear authors, the article is interesting because it deals with a topical issue of great relevance to the scientific community.

  • Thank you.

I make some general comments and specific suggestions that I hope will improve the manuscript:

Check for typos

  • We have gone through the manuscript and corrected what we found. Please see track changes for details.

All bacterial species must be reported in italic font

  • This has now been done. Thank you for reminding us.

Keyword: add Gallibacterium anatis

  • We agree. Thank you for the reminder.

Line 42 - Enrofloxacin (ENR)

Line 45 - Nalidix (NAL)

Line 60 - Minimum inhibitory concentrations (MIC)

  • The three abovementioned abbreviations have now been added, as suggested. Thanks.

Bacterial strains: some additional information on the selected strains would be appropriate. Are they part of already published research? How were they stored? How was the quality of the isolates ensured?

  • As described in the materials and methods section, the strains have been published previously, or are currently under review (Karwanska et al). They have been referenced accordingly. We have not reiterated details on isolation and storage, but this information is available through the appropriate references.

Which laboratory carried out the pivotal analysis of the research?

  • As described in the Author Contributions section, this study has several aspects of analyses within it. As seen from the “Investigation” field, there are contributions from Department of Epizootiology with Exotic Animal and Bird Clinic, Wroclaw University of Environmental and Life Sciences, Poland, as well as from the Department of Veterinary and Animal Sciences at the University of Copenhagen, Denmark. These are mainly strain collection and MIC testing Formal analysis, meaning all subsequent analyses that make up this manuscript was carried out at the University of Copenhagen.

Finally, does the study have any limit?

  • We have included study limitations in the discussion section (lines 342-347)

Reviewer 4 Report

In the current manuscript nothing much has been done from experimental point of view. This paper I believe is not suitable for publication in such a prestigious journal. It can be published in some other journal.

At various places the organism names need to be italicized line line number 300, 334 and 335. And overall the English language need to be revised. 

Author Response

Reviewer 4

Comments and Suggestions for Authors

In the current manuscript nothing much has been done from experimental point of view. This paper I believe is not suitable for publication in such a prestigious journal. It can be published in some other journal.

  • We respectfully disagree. We have obtained phenotypical data, although to a limited extend, and importantly, mechanistic insights into development of antimicrobial resistance, through extensive bioinformatic analyses, ranging from de novo assembly of draft genomes, through full genome annotation and extensive data curation to statistical processing, and structural modelling. In this way, we have been able to combine existing data, new MIC determinations and epidemiological data to form the basis for further, novel examinations of the mechanisms behind the observed development of resistance. That way, swapping pipettes for code, growth media for cloud computing and combining old and new data has been able to provide novel insights.

Comments on the Quality of English Language

At various places the organism names need to be italicized line line number 300, 334 and 335. And overall the English language need to be revised.

  • We have gone through the manuscript and corrected organism names to proper italics. Please see the track changes for full overview of the changes made.

We have also made corrections when needed (typos, double space, etc) to improve grammar. Beyond those, we are quite confident that the level of English used in this manuscript meets the standard of the journal and is appropriate to the intended information.

Round 2

Reviewer 2 Report

The manuscript modified as per the comments and were satisfactory.

Reviewer 4 Report

I still believe this paper is not suitable for this journal as it has more of only bioinformatic work and nothing much interest to the readers. 

Quality of the English language is better but it still needs improvement.